# Impact of heavy precipitation events on pathogen occurrence in estuarine areas of the Puzi River in Taiwan

Yi-Jia Shih[1,2☉], Jung-Sheng Chen[2,3], Yi-Jen Chen[4☉], Pei-Yu Yang[5,6☉], Yi-Jie Kuo[7☉], Tsung-Hsien Chen[8], Bing-Mu Hsu[2,9]*

**1** Fujian Provincial Key Laboratory of Marine Fishery Resources and Eco-environment, Fisheries College, Jimei University, Xiamen, China, **2** Department of Earth and Environmental Sciences, National Chung Cheng University, Chiayi, Taiwan, **3** Department of Medical Research, E-Da Hospital, Kaohsiung, Taiwan, **4** Department of Chest Division, Internal Medicine, Ditmanson Medical Foundation Chiayi Christian Hospital, Chiayi, Taiwan, **5** Department of Laboratory, Show Chwan Memorial Hospital, Changhua, Taiwan, **6** Department of Kinesiology, Health and Leisure, Chienkuo Technology University, Changhua City, Taiwan, **7** Department of Orthopedic Surgery, Wan Fang Hospital, Taipei Medical University, Taipei, Taiwan, **8** Department of Internal Medicine, Ditmanson Medical Foundation Chiayi Christian Hospital, Chiayi, Taiwan, **9** Center for Innovative on Aging Society (CIRAS), National Chung Cheng University, Chiayi, Taiwan

☉ These authors contributed equally to this work.
* bmhsu@ccu.edu.tw

**Data Availability Statement:** The Figs, Raw data and Tables of this study are in the Supporting Information and on Figshare. (https://doi.org/10.6084/m9.figshare.14502780.v1).

## Abstract

Pathogen populations in estuarine areas are dynamic, as they are subject to multiple natural and anthropogenic challenges. Heavy rainfall events bring instability to the aquatic environment in estuaries, causing changes in pathogen populations and increased environmental sanitation and public health concerns. In this study, we investigated the effects of heavy precipitation on the occurrence of pathogens in the Puzi River estuary, which is adjacent to the largest inshore oyster farming area in Taiwan. Our results indicated that *Vibrio parahaemolyticus* and adenovirus were the most frequently detected pathogens in the area. There was a significant difference (Mann-Whitney U test, $p < 0.01$) in water quality parameters, including total coliform, *Escherichia coli*, water temperature, turbidity, salinity, and dissolved oxygen, between groups with and without *V. parahaemolyticus*. In addition, the detection rate was negatively correlated with the average daily rainfall ($r^2 > 0.8$). There was no significant difference between water quality parameters and the presence/absence of adenovirus, but a positive correlation was observed between the average daily rainfall and the detection rate of adenovirus ($r^2 \geq 0.75$). We conclude that heavy precipitation changes estuarine water quality, causing variations in microbial composition, including pathogens. As extreme weather events become more frequent due to climate change, the potential impacts of severe weather events on estuarine environments require further investigation.

**Funding:** This research was supported by the Ministry of Science and Technology of Taiwan (MOST 105-2116-M-194-014), Centers for Disease Control, Taiwan, R.O.C. (MOHW105-CDC-C-114-112601 and 106-CDC-C-114-122601), Show Chwan Health Care System (RD108018), Wan Fang Hospital (108-wf-swf-10) and Ditmanson Medical Foundation Chiayi Christian Hospital-National Chung Cheng University Joint Research Program (R110-25). This research was also supported by the Center for Innovative Research on Aging Society (CIRAS) from the Featured Areas Research Center Program within the framework of the Higher Education Sprout Project by the Ministry of Education (MOE) in Taiwan.

**Competing interests:** The authors have declared that no competing interests exist.

## Introduction

In estuaries, salinity levels change remarkably because of the mixing of tidal flows and the seasonal variability in the amount of freshwater carried by. Steep salinity gradients in estuaries have been shown to result in large changes in bacterial diversity, community structure, and potential growth rates [1–3]. Other factors in estuaries, such as temperature, nutrients, turbidity, and many physical determinants, are known to affect microbial populations with respect to population size, community structure, and the predominant species [1, 3–10]. Many pathogens, including *Vibrio spp.*, *Salmonella spp.*, *Aeromonas spp.*, *Streptococci*, *Bacillus cereus*, *Clostridium difficile*, hepatitis A virus, norovirus, and adenovirus, have been frequently found in estuarine waters [7–9, 11–16]. These pathogens may reach estuaries through the discharge of untreated sewage, wildlife waste, animal farming wastewater, or surface runoff, which could be further affected by the estuarine environment [17–19].

Estuarine waters are rich in organic matter, which makes them an excellent habitat for aquaculture. Some of the most common aquatic species in these environments are euryhaline fish, crustaceans, and mollusks. Aquatic pathogens can cause economic losses due to an increase in disease and mortality in aquaculture species [20–24]. Some aquaculture species are filter feeders, which can cause transmission of pathogens to humans through foodborne infections [25]. In some extreme cases, *Vibrio* organisms such as *Vibrio vulnificus* and *Vibrio cholerae* have been reported to cause illness or lethal septicemia in humans after exposure to pathogen-containing seawater [3, 7, 15, 26]. Therefore, to maintain public health, it is urgent to survey pathogens in estuarine environments.

Seasonal changes affect microbial communities in many aquatic habitats. In estuaries, the communities of bacterioplankton show predictable seasonal patterns that might be associated with salinity [2]. Recently, our group has demonstrated that seasonal changes affect the detection rates and concentrations of human adenovirus in aquatic environments, which is associated with human gastroenteritis. Evidence from previous studies also suggest that the physical determinants and chemical compositions of water bodies might vary seasonally and, consequently, affect microbial communities [27, 28]. Climate change is an important global issue. Due to climate change, the frequency of extreme weather events has increased in recent years. This has caused fluctuations in some physical determinants, including ambient temperature and precipitation, which could interfere with seasonal rhythms in aquatic environments [1, 8]. As the occurrence of extreme weather events becomes more frequent [17, 29, 30], the seasonal pattern of climate determinants has become less predictable, which in turn may affect the overall water quality, biochemical cycles, organism life cycles, and aquatic ecosystems [17, 31, 32].

The effects of climate change on pathogens require further investigation. Research on the impact of climate change on public health has emerged in recent years. For example, various studies have explored potential associations between climate change and waterborne disease infection [26, 32]. Accumulated data have shown that climate change is an important factor associated with the occurrence of certain diseases [33–36]. Nonetheless, most studies have focused on the connection between climate change and pathogenic outbreaks, while the effect of climate on the occurrence of pathogens in estuarine environments is still unclear. According to the 2015 Annual Report by the Taiwanese Council of Agriculture, oyster farming near the estuary of the Puzi River accounts for 43% of the entire oyster production in Taiwan. Precipitation patterns have changed substantially over the past few decades owing to climate change in Taiwan [35, 37]. Whether heavy precipitation affects the occurrence of pathogens in the Puzi River estuary, however, remains unclear. Increased pathogen load is a potential risk to human health from water-transmitted pathogens. For these reasons, we investigated the effects of precipitation events on the occurrence of pathogens in the water bodies of the Puzi River estuary.

Water samples around the Puzi River estuary were collected quarterly to monitor seasonal changes. To screen for dynamic changes in precipitation events, water samples were taken at the same locations on the first, third, eighth, and twelfth days after the end of heavy precipitation events. Commonly found pathogens included *Vibrio spp.*, *C. difficile*, *B. cereus*, norovirus (NoV), hepatitis A virus (HAV), and adenovirus (AdV). These were examined, and the potential effects of intensive precipitation events on microbial water quality in the estuary were also investigated.

## Materials and methods

### Collection of water samples

Water samples were collected from downstream of the Puzi River, at the Dongshing Fishing Port, and from the inshore waters adjacent to the oyster farms. All collections are the publicly accessible sites, which specific permission for collecting water samples was obtained from the central government. The ethics statement of this study had been proved by Centers for Disease Control, Ministry of Health and Welfare of Taiwan central government (MOHW105-CDC-C-114-112601 and 106-CDC-C-114-122601). There is not any endangered/protected land or species were involved in this study. A total of eight sites were investigated during periods of precipitation events; however, only five samples were collected during rain events. The collection of three offshore samples were banned by the Taiwan Coast Guard Administration for safety reasons. Moreover, sample collection was also carried out quarterly once per season from winter (December to February was defined as the winter season) in 2016 to winter in 2017 to monitor seasonal changes in water quality background. The longitude and latitude for each sampling site are illustrated in Fig 1.

To evaluate the impact of heavy precipitation on water quality around the estuary in this study, water samples were collected from the same areas after heavy precipitation events. The precipitation intensity classification is based on the definitions of the Central Weather Bureau in Taiwan: 1) heavy rain, more than 80 mm within 24 h; 2) extremely heavy rain, more than 200

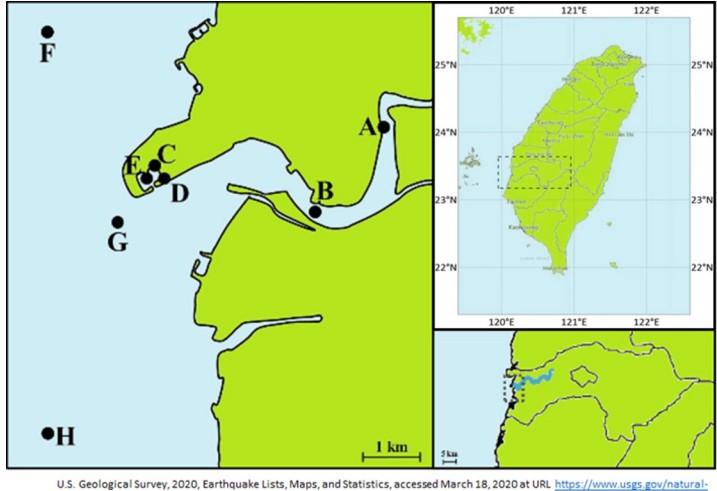

U.S. Geological Survey, 2020, Earthquake Lists, Maps, and Statistics, accessed March 18, 2020 at URL https://www.usgs.gov/natural-hazards/earthquake-hazards/lists-maps-and-statistics

**Fig 1. Sampling locations around the estuarine areas of the Puzi River in Taiwan.** Sites A and B were downstream of the Puzi River, sites C to E were fishing ports, and sites F to H were offshore oyster farms. The geographical coordinates (latitude, longitude) are as follows: A (23.458433, 120.177433), B (23.445306, 120.165667), C (23.452656, 120.138769), D (23.450560, 120.139850), E (23.451032, 120.137538), F (23.473775, 120.120292), G (23.444178, 120.132592), and H (23.410706, 120.120975).

mm within 24 h, or more than 100 mm within 3 h. The dynamic changes in water quality parameters were evaluated on the first, third, eighth, and twelfth days after the events were established. The rainfall data were obtained through the website of the Taiwan Central Weather Bureau Observation Data Inquire System (http://e-service.cwb.gov.tw/HistoryDataQuery/index.jsp). The weather parameters of the two events were documented in Fig 2. One was classified as heavy rain, which occurred in the East Asian Rainy season on June 12. The other rainfall event was classified as extremely heavy rain, which appeared after Typhoon Nepartak on July 10 (see Table 1). In both events, the precipitation tapered off after the first couple of days. Although some rainfall occurred afterwards, the precipitation was no more than 5% of the first three days.

The Figs of this study have been placed in website **Figshare** (https://doi.org/10.6084/m9.figshare.14502780).

## Water quality examination

About 3-L of water sample was collected for testing during each sampling. The samples were stored at 4°C and transported to the laboratory within 6 h. Water samples were subjected to a spectrum of quality tests, including temperature, pH, dissolved oxygen, turbidity, and microbial analyses. The water temperature, pH, and dissolved oxygen were determined using a portable multi-parameter water quality meter (HI9828, Hanna Instruments, USA) immediately after sample collection. Turbidity was determined using a ratio turbidimeter (Waterproof Portable TN100, Eutech Instruments, Singapore). For microbial analysis, heterotrophic plate count was performed using the spread plate method with m-HPC agar according to the standard protocol (Methods 9215C) [38]. Total coliform was measured by membrane filtration and differential medium as described in the standard method for water and wastewater

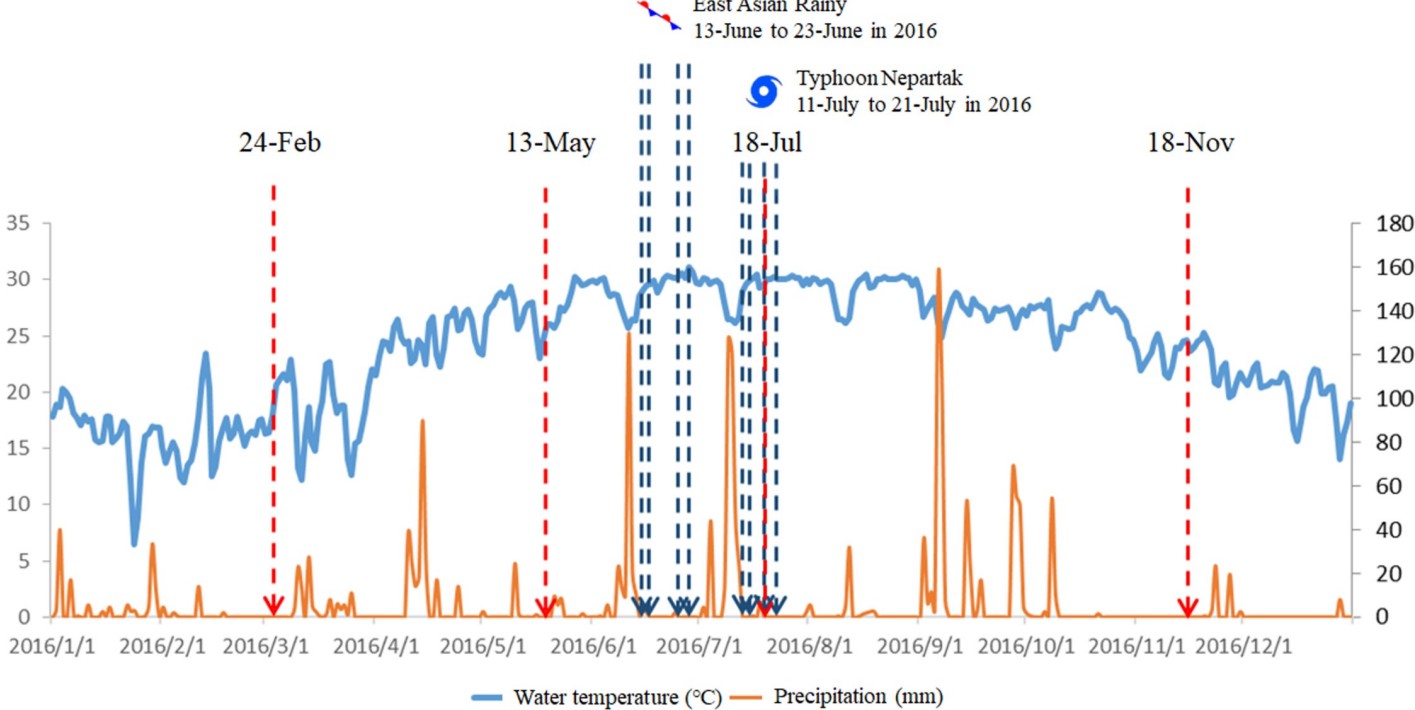

**Fig 2. Timeline of weather parameters around estuarine areas of the Puzi River and the sampling dates.** The dashed red arrow bar indicates the date for seasonal sampling. The dashed blue arrow bar indicates the sampling dates during rainfall events.

**Table 1. Seasonal water quality parameters around the estuarine areas of the Puzi River.**

| Water quality parameters | Sampling dates | | | | |
|---|---|---|---|---|---|
| | 2016/2/24 (n = 8) | 2016/5/13 (n = 8) | 2016/7/18 (n = 8) | 2016/11/18 (n = 8) | 2017/2/8 (n = 8) |
| Heterotrophic plate count (CFU/mL) | 354±1,387 | 12,366±5,218 | 38,912±11,310 | 3,566±1,623 | 335±1421 |
| Total Coliform (CFU/100mL) | 254±511 | 409±502 | 631±2,023 | 280±2662 | 224±483 |
| *Escherichia coli* (CFU/100mL) | 23±79 | 48±40 | 130±255 | 20±22 | 22±83 |
| pH | 8.6±0.6 | 8.0±0.2 | 7.9±0.1 | 7.6±0.2 | 8.4±0.3 |
| Turbidity (NTU) | 15.2±20.8 | 11.9±28.2 | 39.2±34.6 | 27.3±27.1 | 16.7±19.8 |
| Salinity (%) | 29.9±1.5 | 27.5±1.9 | 9.3±6.8 | 21.6±12.2 | 20.7±5.3 |
| Dissolved oxygen (mg/L) | 7.0±0.4 | 4.8±0.4 | 4.5±0.9 | 5.7±0.8 | 6.9±0.3 |
| Average water temperature (˚C) | 18.2±0.9 | 29.9±0.7 | 28.9±0.4 | 27.0±1.2 | 20.2±1.2 |

examination (Methods 9222 B) [38]. The quality control and quality assurance methods in this study were carried out as described in our previous report [27].

## Sample pretreatment for microbial analyses

For detection of specific microbial pathogens, a water sample (300 mL) was filtered through a 47 mm GN-6 membrane (Pall, Mexico City, Mexico) with a pore size of 0.45 μm, which was held by a stainless filter holder. Subsequently, the membranes were used for the enrichment of microbes, as described in the next section. For the detection of the virus, the virus was recovered by the method described in our earlier studies [27, 39, 40]. In short, each water sample (1 L) was filtered through 47 mm GN-6 membranes (Pall, Mexico City, Mexico) with a pore size of 0.45 μm. Viruses attached to the filter membrane were scraped manually into 50 mL of sterilized phosphate-buffered saline (PBS). Subsequently, the eluent was transferred into two conical centrifuge tubes (50 mL each) and centrifuged at 2600×*g* for 30 min (KUBOTA Model 2420 Compact Tabletop Centrifuge, Japan). The pellet was resuspended in PBS (5 mL) for total viral DNA extraction at 4˚C.

## Enrichment of bacterial isolates

For the detection of various pathogens, water samples were enriched according to their optimal growth conditions. For *Vibrio spp.*, the water samples were enriched with alkaline peptone water (APW) and selectively cultured on CHROMagar™ *Vibrio* and thiosulfate-citrate-bile salts-sucrose agar (TCBSA). For *V. parahaemolyticus* and *V. cholerae*, samples were incubated at 37˚C for 24 h. For *V. vulnificus*, samples were incubated at 30˚C for 24 h. For *Clostridium difficile*, samples were enriched with cycloserine-cefoxitin fructose broth (CCFB) and selectively cultured on CHROMagar™ *Clostridium difficile* and cycloserine-cefoxitin fructose agar (CCFA), and incubated at 37˚C for 24 h to 48 h under anaerobic conditions. For *B. cereus*, samples were enriched with trypticase soy polymyxin broth (TSPB), selectively cultured on CHROMagar™ *Bacillus cereus* and mannitol egg yolk polymyxin agar (MEYP), and incubated at 37˚C for 24 h to 48 h.

## Microbial DNA extraction and identification

For bacterial detection, DNA was extracted from the second enrichment broth (1 mL) using the MagPurix bacteria DNA Extraction Kit ZP02006. For virus detection, DNA was extracted from the pellet as described in the previous section. Total viral DNA/RNA was extracted using the Viral Nucleic Acid Extraction Kit ZP02003. MagPurix 12s Automated Nucleic Acid Purification System (Zinexts Life Science Corp., Taiwan) was employed for DNA extraction. PCR

was carried out on the final DNA extraction product (100 μL) to confirm the species of pathogens. The experimental conditions of primer sequences, positive controls of pathogens, and their references are summarized in Table 2.

## Statistical analysis

The Mann-Whitney U test was performed to examine the correlation between physical and biological water quality parameters and the occurrence of pathogens after a precipitation event, and the correlation between cumulative rainfall of a precipitation event and the occurrence of pathogens. The average daily precipitation is calculated as follows:

$$\text{Average daily precipitation } = \frac{(\text{RE} + \text{RN})}{(\text{RED} + \text{RND})}, \text{ where}$$

RE: Cumulative precipitation depth (mm/24 h) during the event;

RN: Cumulative precipitation depth (mm/24 h) during the sampling period (days 1 to N);

RED: Duration of the precipitation event in days;

RND: Cumulative days counted from the first day of the precipitation event to the sampling day.

The statistical analyses were performed using Paleontological Statistics Version 3.14 (PAST, University of Oslo, Norway) and SigmaPlot V.10.0. software (Systat Software Inc., US). Differences were considered significant when the p value was ≤ 0.05.

The Figs, Raw data and Tables of this study have been placed in website **Figshare** (https://doi.org/10.6084/m9.figshare.14502780).

## Results

### Pathogen detection rates of seasonal levels and after rainfall events

Of all the pathogens examined in this study, *V. parahaemolyticus* and AdVs were the most frequently detected bacteria and viruses, with detection rates of 90% (29/32) and 25% (8/32), respectively (Table 3). Since there was no statistically significant difference in detection rates of *V. parahaemolyticus* and AdVs according to seasonal changes, the data from 4 seasons were combined to compare to the detection rates after precipitation events. After heavy rainfall, the detection rate of *V. parahaemolyticus* immediately dropped to 40% and then gradually increased to the original high level. On the other hand, the detection rate of AdVs reached 80% and decreased to 12.5% after 12 days. As for the other pathogens tested in this study, i.e., *V. cholerae*, *V. vulnificus*, *B. cereus*, *C. difficile*, NoV, and HAV, they were detected occasionally during quarterly tests and not affected by precipitation events.

### Correlation between precipitation and the occurrence of pathogens

Fig 3 shows the correlation between the average daily precipitation and detection rates of *V. parahaemolyticus* and AdVs. In both cases, the average daily precipitation decreased with time from the first day of the precipitation event to the twelfth sampling day, while the detection rates for *V. parahaemolyticus* increased in the same period (Fig 3A and 3B). There was a negative correlation between precipitation and detection rates for *V. parahaemolyticus* ($r^2 > 0.8$). The extremely heavy rain did not increase the slope value of the correlation curve between the detection rate of *V. parahaemolyticus* and daily precipitation. The detection rate of *V.*

**Table 2. The primer sequences and positive controls of pathogens.**

| Pathogens | Target gene | Size | Sequence (5′ to 3′) | Positive control | References for reaction materials and PCR condition |
|---|---|---|---|---|---|
| *Bacillus cereus* | *Bal* | 533 | BalF: 5′-TGCAACTGTATTAGCACAAGCT-3′ | *B. cereus*, ATCC 11778 | [41] |
| | | | BalR: 5′-TACCACGAAGTTTGTTCACTACT-3′ | | |
| *Clostridium difficile* | *tcdB* | 632 | tcdA-F 5′-GTATGGATAGGTGGAGAAGTCAGTG-3′ | *C. difficile*, ATCC 9689 | [42] |
| | | | tcdA-R 5′-CGGTCTAGTCCAATAGAGCTAGGTC-3′ | | |
| | *tcdA* | 441 | tcdB-F 5′-GAAGATTTAGGAAATGAAGAAGGTGA-3′ | | |
| | | | tcdB-R 5′-AACCACTATATTCAACTGCTTGTCC-3′ | | |
| | *cdtA* | 260 | cdtA-F 5′-ATGCACAAGACTTACAAAGCTATAGTG-3′ | | |
| | | | cdtA-R 5′-CGAGAATTTGCTTCTATTTGATAATC-3′ | | |
| | *cdtB* | 179 | cdtB-F 5′-ATTGGCAATAATCTATCTCCTGGA-3′ | | |
| | | | cdtB-R 5′-CCAAAATTTCCACTTACTTGTGTTG-3′ | | |
| *Vibrio cholera* | ompW | 427 | VCompW-F: 5′-CACCAAGAAGGTGACTTTATTGTG-3′ | *V. cholera*, ATCC 14035 | [43] |
| | | | VCompW-R: 5′-CGTTAGCAGCAAGTCCCCAT-3′ | | |
| *V. parahaemolyticus* | Collagenase | 271 | VPC-F: 5′-GAAAGTTGAACATCATCAGCACGA-3′ | *V. parahaemolyticus*, ATCC 17802 | [44] |
| | | | VPC-R: 5′-GGTCAGAATCAAACGCCG-3′ | | |
| *V. vulnificus* | vvhA | 505 | FDAvvhA-F: 5′ -CCGCGGTACAGGTTGGCGCA-3′ | *V. vulnificus*, ATCC 27562 | [43] |
| | | | FDAvvhA-R: 5′-CGCCACCCACTTTCGGGCC-3′ | | |
| Hepatitis A virus | 3D | 412 | HAV1: 5′-TTTGGTTGGATGAAAATGGTT-3′ | Internal controls: | [45] |
| | | | HAV4: 5′-ATTCTACCTGCTTCTCTAATC-3′ | | |
| | | 233 | HAV2: 5′-CAACCTGTCCAAAAGATGAAT-3′ | | |
| | | | HAV3: 5′-ACCAACATCTCCGAATCTTA-3′ | | |
| Norovirus | ORF2 | | Group I, GI: | | [46] |
| | | | COG1F: 5′-CGYTGGATGCGNTTYCATGA-3′ | | |
| | | 377 | G1-SKF: 5′-CTGCCCGAATTYGTAAATGA-3′ | | |
| | | 330 | G1-SKR: 5′-CCAACCCARCCATTRTACA-3′ | | |
| | | 386 | Group II GII: | Bacteria universal 16S rDNA PCR for avoiding false negative results | |
| | | 344 | COG2F: 5′-CARGARBCNATGTTYAGRTGGATGAG-3′ | | |
| | | | G2-SKF: 5′- CNTGGGAGGGCGATCGCAA -3′ | | |
| | | | G2-SKR: 5′- CCRCCNGCATRHCCRTTRTACAT -3′ | | |
| Adenovirus | Hexon neHexon | 301 | Hex: | | [47] |
| | | | Hex1 deg F: 5′-GCCSCARTGGKCWTACATGCACATC-3′ | | |
| | | | Hex2 deg R:5′-CAGCACSCCICGRATGTCAAA-3′ | | |
| | | 171 | neHex: | | |
| | | | neHex 3deg F: 5′-GCCCGYGCMACIGAIACSTACTTC-3′ | | |
| | | | neHex 4deg R: 5′-CCYACRGCCAGIGTRWAICGMRCYTTGTA-3′ | | |

*parahaemolyticus* in quarterly tests was about 90% to 100% in the estuary of the Puzi River. In contrast, a positive correlation was found between the detection rates of AdVs and precipitation (Fig 3C and 3D, $r^2 \geq 0.75$). The quarterly detection rate of AdVs was approximately 25% in estuary areas. After both precipitation events, the detection rates of AdVs rose to 60%–80%

**Table 3. The presence of pathogens in estuarine areas of the Puzi River after rainfall.**

| | Days after heavy precipitation (by East Asian Rainy) | | | | Days after extremely heavy precipitation (by Typhoon Nepartak) | | | | Seasonal background |
|---|---|---|---|---|---|---|---|---|---|
| | Day 1 | Day 3 | Day 8 | Day 12 | Day 1 | Day 3 | Day 8 | Day 12 | n = 32 |
| | (n = 5) | (n = 8) | (n = 8) | (n = 8) | (n = 8) | (n = 8) | (n = 8) | (n = 8) | |
| *Vibrio parahaemolyticus** | 2 | 6 | 6 | 8 | 6 | 7 | 7 | 8 | 29 |
| *Vibrio cholera* | 0 | 0 | 0 | 0 | 0 | 0 | 1 | 1 | 4 |
| *Vibrio vulnificus* | 0 | 0 | 0 | 0 | 2 | 1 | 0 | 0 | 0 |
| *Bacillus cereus* | 0 | 0 | 0 | 0 | 0 | 0 | 2 | 1 | 1 |
| *Clostridium difficile* | 0 | 0 | 0 | 0 | 0 | 0 | 2 | 1 | 0 |
| Adenovirus* | 4 | 5 | 5 | 3 | 5 | 5 | 3 | 1 | 8 |
| Norovirus (G1/G2) | 0 | 0 | 1 | 1 | 0 | 0 | 0 | 0 | 2 |
| HAV (hepatitis A virus) | 0 | 0 | 0 | 0 | 0 | 0 | 0 | 0 | 0 |

*indicates dominant pathogen.

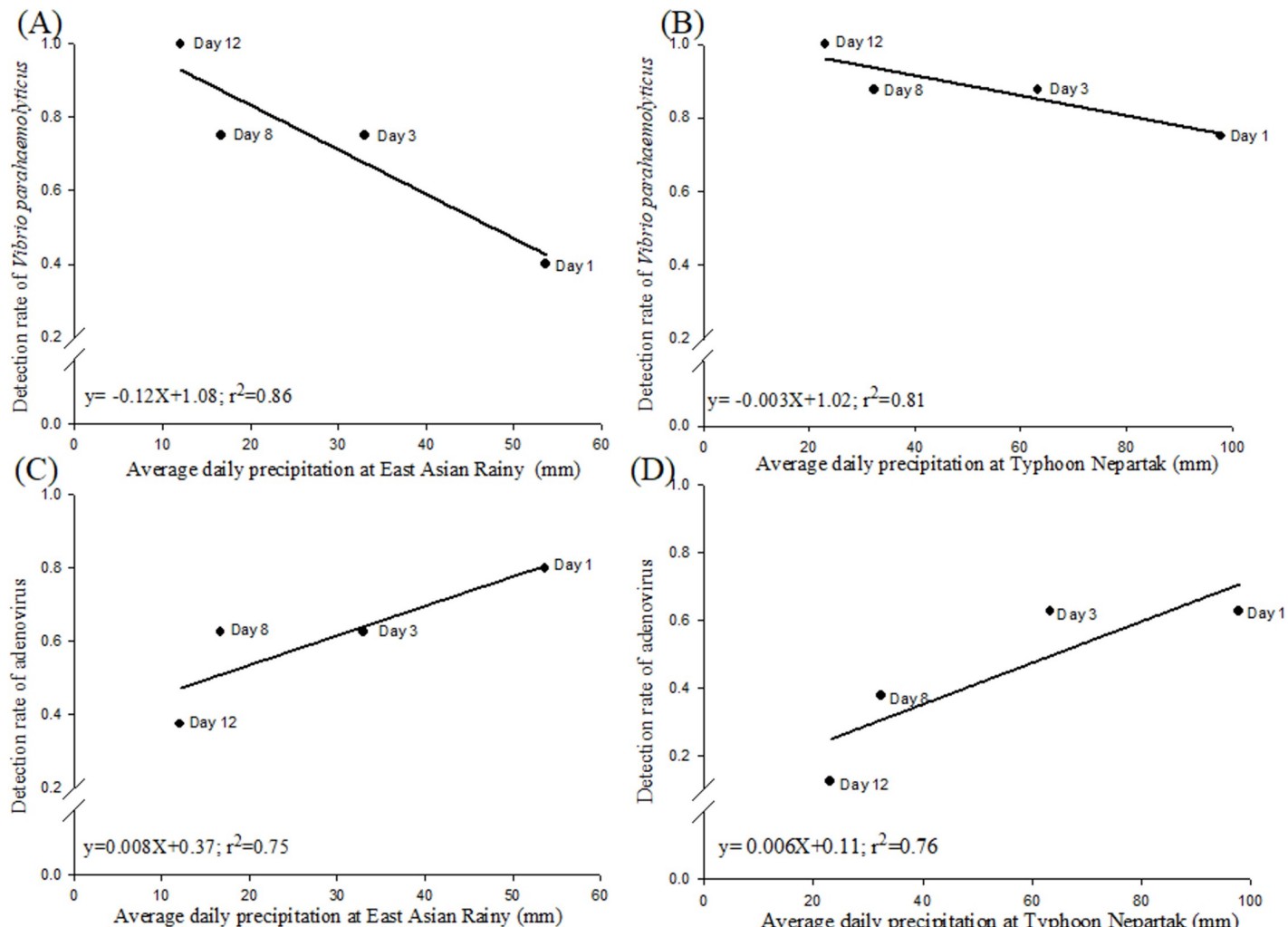

**Fig 3. Correlation between the detection rates of pathogens and the averages of daily precipitation.**

and returned to the original levels within 12 days after the events. A higher precipitation level did not further enhance the detection rate of AdVs.

## Differences in water quality parameters and the occurrence of pathogens

Table 3 presents the data on water quality parameters, including temperature, pH, salinity, dissolved oxygen, turbidity, and microbial indicators, from quarterly samples. Seasonal changes were observed in environmental parameters, such as temperature, dissolved oxygen, and several biological parameters. It should be pointed out that the third quarterly samples were taken on the eighth day after extremely heavy precipitation events, which could have interfered with the regular pattern.

The effects of precipitation events on environmental parameters are summarized in Tables 4 and 5. The parameters of heterotrophic plate count (HPC), total coliform, and *E. coli* increased significantly immediately after the occurrence of heavy rainfall and gradually decreased. In addition, rainfall causes an increase in turbidity and dissolved oxygen in water, which decreases over time afterwards. Rainfall also dropped the water temperature by 3 degrees and decreased salinity drastically. Rainfall has little effect on the changes in pH levels in estuarine areas. The average water quality parameters variation by different sampling sites was shown on S1–S3 Tables.

Table 6 summarizes the results of statistical significance in water quality parameters between the groups with or without the occurrence of *V. parahaemolyticus* using the Mann-Whitney U test. The occurrence of *V. parahaemolyticus* was associated with higher salinity, higher temperature, lower turbidity, and lower numbers of total coliform and *E. coli*. In contrast, the occurrence of AdVs in the water samples after precipitation events was not associated with any of the water quality parameters in this study (Table 7).

## Discussion

As inshore farming of oysters around the Puzi River estuary is the primary source of oysters in Taiwan, and human activities and livestock farms are located around the Puzi River basin [48], the wastewater from the river is an important contributor to the water quality of the estuary. Therefore, it is necessary to monitor the dynamics of microbial communities in these areas. Microbial communities in estuarine environments are easily affected by weather, river flow, tidal cycles, sea temperature, and salinity [1, 3, 10]. As severe weather events are likely to affect pathogens due to changes in environmental factors, they pose potential threats to human

**Table 4. Water quality parameters around the estuarine areas of the Puzi River after heavy precipitation.**

| Sampling events | Days after heavy precipitation (by East Asian Rainy) | | | |
|---|---|---|---|---|
| Water quality parameters | Day 1 (n = 5) | Day 3 (n = 8) | Day 8 (n = 8) | Day 12 (n = 8) |
| Heterotrophic plate count (CFU/mL) | 86,093±20,618 | 63,142±12,910 | 16,366±7,819 | 3,378±1,720 |
| Total Coliform (CFU/100mL) | 9,700±2,953 | 5,351±1,507 | 941±645 | 78±75 |
| *Escherichia coli* (CFU/100mL) | 557±222 | 253±106 | 188±19 | 1±1 |
| pH | 7.6±0.1 | 7.6±0 | 7.7±0.1 | 7.9±0 |
| Turbidity (NTU) | 78.5±32.4 | 25.8±6.7 | 20.5±11.6 | 1.8±0.4 |
| Salinity (PSU) | 2.9±1.1 | 8.5±2.1 | 11.8±2.7 | 15.7±2.2 |
| Dissolved oxygen (mg/L) | 5.3±0.1 | 4.7±0.1 | 4.2±0.1 | 3.8±0.2 |
| Average water temperature (˚C) | 26.4±0.1 | 28.0±0.2 | 28.9±0.3 | 29.3±3.1 |
| Cumulative precipitation (mm) | 161 | 165 | 167 | 169 |
| Average daily precipitation (mm) | 53.7 | 33.0 | 16.7 | 12.1 |

**Table 5. Water quality parameters around the estuarine areas of the Puzi River after extreme heavy precipitation.**

| Sampling events | Days after extreme heavy precipitation (by Typhoon Nepartak) | | | |
|---|---|---|---|---|
| Water quality parameters | Day 1 (n = 8) | Day 3 (n = 8) | Day 8 (n = 8) | Day 12 (n = 8) |
| Heterotrophic plate count (CFU/mL) | 314,381±180,326 | 716,950±471,663 | 38,912±41,879 | 27,383±22,439 |
| Total Coliform (CFU/100mL) | 2,390±1,639 | 1,149±823 | 631±708 | 52±87 |
| *Escherichia coli* (CFU/100mL) | 133±128 | 81±33 | 130±95 | 24±21 |
| pH | 8.1±0.1 | 7.7±0.1 | 7.9±0.1 | 7.8±0.1 |
| Turbidity (NTU) | 55.3±19.6 | 30.7±14.7 | 30.2±11.1 | 11.9±4.6 |
| Salinity (PSU) | 2.9±1.0 | 1.1±0.6 | 9.3±3.1 | 15.3±1.4 |
| Dissolved oxygen (mg/L) | 7.4±0.6 | 4.5±0.2 | 4.5±0.6 | 4.8±0.4 |
| Average water temperature (˚C) | 26.1±0.1 | 27.2±0.2 | 28.9±0.3 | 29.2±0.3 |
| Cumulative precipitation (mm) | 294 | 317 | 317 | 324 |
| Average daily precipitation (mm) | 97.8 | 63.3 | 32.4 | 23.1 |

health, though the extent of this threat has not been determined. *Vibrio* spp. and AdV were the most frequently detected pathogens in this study. Previous studies have shown that halophilic *Vibrio* species are a major group of pathogens in estuarine environments [7, 8, 22]. As confirmed by PCR and DNA sequencing, we demonstrated that *V. parahaemolyticus* and AdVs (human adenovirus 41 [HAdV-41] and porcine adenovirus 5 [PAdV-5]) were the predominant pathogens in the Puzi River estuary.

Previous studies have demonstrated that *V. parahaemolyticus* is a serious pathogen that causes foodborne illness in many outbreaks [24, 35, 41, 42]. This bacterium survives in high-temperature marine environments and is frequently isolated from a variety of seafood, including oysters, mollusks, and crustaceans. *Vibrio* species can also cause economic losses in aquaculture by interfering with the growth of mollusks and crustaceans [24]. After the events of heavy precipitation, *Eichhornia crassipes*, commonly known as water hyacinth, which grows in the Puzi River, washed down and piled up separately around the estuary. We found that the water temperature, turbidity, and salinity of the estuary areas were affected by heavy rainfall. Our data indicated that the occurrence of *V. parahaemolyticus* was significantly associated with several parameters such as temperature, turbidity, and salinity according to the results of the Mann-Whitney U test. Previous studies have shown that the occurrence and abundance of halophilic *Vibrio* are affected by changes in salinity, water temperature, and turbidity, which is consistent with the current study [8, 9, 35, 42, 43]. Interestingly, the heterotrophic plate count

**Table 6. Comparisons between water quality parameters and the presence of *Vibrio parahaemolyticus*.**

| Water quality parameters | Mann-Whitney *U* test | VP- Positive (n = 50) | | | VP-Negative (n = 11) | | |
|---|---|---|---|---|---|---|---|
| | | Median | $Q_1$ | $Q_3$ | Median | $Q_1$ | $Q_3$ |
| Heterotrophic plate count (CFU/mL) | $p = 0.68$ | 78,000 | 14,053 | 304,867 | 61,500 | 23,883 | 91,133 |
| Total Coliform (CFU/100mL)* | $p = 0.003$ | 347 | 16 | 3,309 | 4,964 | 2,591 | 11,512 |
| *Escherichia coli*(CFU/100mL)* | $p = 0.002$ | 32 | 0 | 125 | 415 | 166 | 700 |
| Water temperature (˚C)* | $p = 0.007$ | 31.0 | 30.1 | 32.3 | 29.0 | 26.5 | 29.9 |
| pH | $p = 0.13$ | 7.8 | 7.6 | 8.0 | 7.6 | 7.5 | 7.8 |
| Turbidity (NTU)* | $p = 0.001$ | 14.0 | 3.5 | 20.5 | 69.1 | 33.5 | 131.2 |
| Salinity (PSU)* | $p<0.001$ | 11.7 | 3.3 | 16.7 | 0.7 | 0.1 | 0.9 |
| Dissolved oxygen (mg/L) | $p = 0.08$ | 4.6 | 4.0 | 5.2 | 5.2 | 4.7 | 6.0 |

*indicates $p<0.05$; VP for *Vibrio parahaemolyticus*.

Q1 for first quartile; Q3 for third quartile.

**Table 7. Comparisons between water quality parameters and the presence of adenovirus.**

| Water quality parameters | Mann-Whitney $U$ test | AdVs- Positive (n = 31) | | | AdVs-Negative (n = 30) | | |
|---|---|---|---|---|---|---|---|
| | | Median | $Q_1$ | $Q_3$ | Median | $Q_1$ | $Q_3$ |
| Heterotrophic plate count (CFU/mL) | $p = 0.17$ | 95,733 | 22,998 | 304,217 | 25,383 | 2,418 | 255,000 |
| Total Coliform (CFU/100mL) | $p = 0.21$ | 2,399 | 38 | 5,186 | 453 | 42 | 2,297 |
| *Escherichia coli*(CFU/100mL) | $p = 0.25$ | 86 | 0 | 299 | 34 | 0 | 96 |
| Water temperature (°C) | $p = 0.81$ | 30.8 | 29.8 | 32.2 | 30.6 | 29.9 | 31.9 |
| pH | $p = 0.47$ | 7.8 | 7.6 | 8.0 | 7.7 | 7.5 | 8.0 |
| Turbidity (NTU) | $p = 0.70$ | 18.6 | 5.8 | 33.1 | 15.2 | 3.7 | 32.0 |
| Salinity (PSU) | $p = 0.94$ | 4.8 | 1.3 | 14.2 | 9.1 | 0.9 | 15.3 |
| Dissolved oxygen (mg/L) | $p = 0.11$ | 4.8 | 4.4 | 5.5 | 4.5 | 4.0 | 5.1 |

*indicates $p < 0.05$; AdVs for adenovirus.

Q1 for first quartile; Q3 for third quartile.

after the typhoon with extremely heavy rainfall was maintained at a high level for almost 2weeks, causing the issue of a warning to human health. It is important to survey whether there are unknown pathogens that have not yet been revealed in this study following events of extremely heavy precipitation.

AdVs are one of the most prevalent waterborne viruses, and they may be easily transmitted through water [14, 36, 44]. In this study, AdVs were found to be the most abundant DNA virus in water samples after heavy precipitation events. There was a positive correlation between the occurrence of adenovirus and the average daily precipitation; however, there was no significant difference between the occurrence of AdVs and any of the water quality parameters that were assayed in this study. It is plausible that the heavy precipitation carried the AdVs from polluted water upstream of the Puzi River and from contaminated adjacent urban areas to the estuary areas, and consequently elevated the AdVs detection rates. AdVs are highly contagious, and large numbers of AdVs can be excreted in human feces or urine from infected individuals [14, 28, 45, 46]. As one of the major origins of AdVs is human feces, AdVs have been proposed as an indicator of human fecal pollution [44, 46, 47]. In this study, the detection rate for AdVs increased significantly straightaway after rainfall events and returned to seasonal levels in 12 days. The pattern suggested that the AdVs gathered by heavy precipitation were run off continuously by water upstream. These data implied that the risk of AdVs infections increased in human and aquaculture species due to exposure to high levels of the virus in estuarine areas after heavy and extremely heavy precipitation events.

In this study, we revealed that the detection rates of two important pathogens were affected by rainfall events in opposite patterns. We proposed three possible underlying mechanisms that may influence the changes in organisms in estuarine areas after precipitation events. First, heavy and extremely heavy precipitation changed the environmental parameters, which may affect the advantages or disadvantages of the survival of predominant pathogens. In this study, we showed that the detection rates of *V. parahaemolyticus* decreased since the environmental conditions did not favor the growth of the organism. Second, precipitation events changed the detection rates of pathogens by carrying them into certain habitats and altering their occurrences. In this study, we found that the presence of adenovirus correlated only with the average daily precipitation, not the water quality parameters, suggesting that rainfall can bring adenovirus to the estuary. Third, heavy precipitation can promote the growth of microbes because heavy precipitation drastically elevates the heterotrophic plate counts. Moreover, the extent of microbial growth was enhanced even further in the extremely heavy precipitation event,

suggesting that the intensity of precipitation played a role in changing the balance of microbial communities. A natural habitat may take a longer time to recover when it experiences extremely heavy events.

Current climate change models suggest that global warming may have altered the frequency and intensity of precipitation and extreme weather events [30, 48]. Changes in precipitation patterns threaten the ecosystem, aquaculture, and other forms of human activities [17, 26, 37]. Environmental changes from extreme weather events may also have direct or indirect impacts on marine microorganisms, especially in estuarine areas [1, 3, 7, 10, 13]. Historical climate data in Taiwan have suggested an increased incidence and intensity of extreme weather events over the past few years [37]. More studies on the changes in microbial communities are needed to reduce the threats of these events.

## Conclusions

The correlation between the presence of commonly detected pathogens and the dynamic changes in water quality and microbial indices after heavy precipitation were investigated around the Puzi River estuary. We found that *V. parahaemolyticus* and AdVs were the most frequently detected pathogens in the estuarine environment after heavy precipitation. The detection rate of *V. parahaemolyticus* decreased immediately after heavy precipitation, followed by slowly returning to the original levels. On the other hand, the detection rates of AdVs surged to a high level after the rainfall events, followed by a gradual decrease to seasonal levels in 12 days. Microbial indices such as *E. coli*, total coliform, and heterotrophic plate count were elevated drastically by precipitation, suggesting that a favorable environment for the growth of microbes is created after rainfall events. As extreme weather events have become more frequent in recent years, more investigations are needed to survey the relative abundance of microbial communities after heavy precipitation events and to identify the pathogens of potential threats that have not yet emerged.

## Supporting information

**S1 Table. Water quality parameters variation of Puzi River (site A-B) after rainfall.**
(DOCX)

**S2 Table. Water quality parameters variation of Dongshing Fishing Port (site C-E) after rainfall.**
(DOCX)

**S3 Table. Water quality parameters variation of oyster farms (site F-H) after rainfall.**
(DOCX)

## Author Contributions

**Conceptualization:** Yi-Jia Shih, Jung-Sheng Chen, Yi-Jen Chen, Bing-Mu Hsu.

**Data curation:** Yi-Jia Shih, Yi-Jen Chen, Pei-Yu Yang, Yi-Jie Kuo.

**Funding acquisition:** Yi-Jen Chen, Pei-Yu Yang, Yi-Jie Kuo, Bing-Mu Hsu.

**Investigation:** Jung-Sheng Chen, Yi-Jie Kuo, Tsung-Hsien Chen.

**Methodology:** Yi-Jia Shih, Pei-Yu Yang, Tsung-Hsien Chen.

**Software:** Yi-Jia Shih, Jung-Sheng Chen.

**Supervision:** Bing-Mu Hsu.

**Validation:** Yi-Jen Chen, Pei-Yu Yang, Yi-Jie Kuo.

**Visualization:** Jung-Sheng Chen, Bing-Mu Hsu.

**Writing – original draft:** Yi-Jia Shih, Yi-Jen Chen, Pei-Yu Yang, Yi-Jie Kuo.

**Writing – review & editing:** Jung-Sheng Chen, Bing-Mu Hsu.

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
