## [Decision Letter · Decision Letter 0]

12 Feb 2021

PONE-D-20-34257

Impact of heavy precipitation on pathogen occurrence in estuary with oyster farming

PLOS ONE

Dear Dr. Hsu,

Thank you for submitting your manuscript to PLOS ONE. After careful consideration, we feel that it has merit but does not fully meet PLOS ONE’s publication criteria as it currently stands. Therefore, we invite you to submit a revised version of the manuscript that addresses the points raised during the review process.

Please respond to all of the reviewers comments.  In particular, revise the title to be less ambiguous and ensure that all data are fully available in a public repository.   Additionally, because this manuscript involves salt-dependent bacteria in an estuarine system, it is not suitable for the call for Freshwater Ecology at PLOS ONE.  Please consider submitting as a stand-alone paper.  

We look forward to receiving your revised manuscript.

Kind regards,

Brett Austin Froelich, Ph.D

Academic Editor

PLOS ONE

Journal Requirements:

2. Please amend either the title on the online submission form (via Edit Submission) or the title in the manuscript so that they are identical.

4.We note that Figure(s) 1 in your submission contain map images which may be copyrighted. All PLOS content is published under the Creative Commons Attribution License (CC BY 4.0), which means that the manuscript, images, and Supporting Information files will be freely available online, and any third party is permitted to access, download, copy, distribute, and use these materials in any way, even commercially, with proper attribution. For these reasons, we cannot publish previously copyrighted maps or satellite images created using proprietary data, such as Google software (Google Maps, Street View, and Earth). For more information, see our copyright guidelines: http://journals.plos.org/plosone/s/licenses-and-copyright.

a) You may seek permission from the original copyright holder of Figure(s) 1 to publish the content specifically under the CC BY 4.0 license. 

Reviewers' comments:

Reviewer's Responses to Questions

**Comments to the Author**

1. Is the manuscript technically sound, and do the data support the conclusions?

Reviewer #1: Partly

2. Has the statistical analysis been performed appropriately and rigorously? 

Reviewer #1: Yes

3. Have the authors made all data underlying the findings in their manuscript fully available?

Reviewer #1: No

4. Is the manuscript presented in an intelligible fashion and written in standard English?

Reviewer #1: Yes

5. Review Comments to the Author

Reviewer #1: Manuscript No.: PONE-D-20-34257

Title: Impact of heavy precipitation on pathogen occurrence in estuary with oyster farming

Authors: YI-JIA SHIH et al

Overview: The paper investigates the impact of heavy rainfall events on the bacterial and viral load in an estuary. The authors sampled at 8 different points in the estuary. Seasonal samples were collected on a roughly 3 month cycle presumably to provide a baseline. Additional sampling was conducted after a single heavy rainfall event (>80mm/24 hours) and a single very heavy rainfall event (>200mm/24 hours). Water samples were screened for basic chemistry and pathogen load, both bacterial and viral. The authors report variations in pathogenic Vibrio sp., and both adenovirus and norovirus.

General comments: While the study is interesting there appear to be a number of flaws in the experimental design. The data presented is an average of 8 relatively dispersed sample sites. There is no attempt to show how each of the sites varies. Since some of the sites are a kilometer up the estuary while others are offshore, there may be substantial variation in water chemistry and pathogen load that is not being reported. There is no discussion of the hydrography/oceanography of the sample area. Some of the errors indicate substantial variability on the sample data

Title: The title is misleading. The inclusion of oyster farming might suggest that the authors investigated pathogen load in oysters, which they did not. In fact, the relevance to oyster farming is somewhat tenuous. I strongly recommend the reference to an oyster farming region is removed.

Abstract: A little long, but relevant

Introduction: Appropriate. The references to the previous work (bibliography number 27 and 28) of the authors on line 67-69 should be cited with this sentence

Methods: The methodology is somewhat confusing. There are a number of issues that need to be clarified.

1. The locations of the sampling points is somewhat confusing in relation to the data collected. The first section indicates samples were collected adjacent to oyster farms. But no further information is given. Which samples? Where are the oyster farms?

2. The first section indicates that ‘seasonal’ samples were collected quarterly. However this is untrue according to the presented data. Seasonal samples were collected in February, May, July and November. This is not quarterly.

3. The water quality data and pathogen load data seems to be a mean of all 8 sample points. These points are reasonably distributed both in estuary and the surrounding coast. Some of the sample point appear to be 2 or 3 kilometers offshore. Is it reasonable to average all these points? Some of the data points have large error.

4. Day 1 sampling for the ‘heavy rain’ even is an average of only 5 samples. There is no explanation or indication which sites were excluded.

5. References listed in table 2 and not in the same format as the rest of the manuscript and are NOT listed in the bibliography.

6. I would expect the seasonal sampling to have run for a full 12 months, or more, to ensure that a complete seasonal cycle is evaluated. That would mean that 5 sample sets should have been collected, including a February set in year 2.

Results: Reasonably appropriate. A number of the tables and figures need to have more descriptive titles and legends. For instance in table 3, both Vibrio parahaemolyticus and adenovirus have an ‘asterisk’ but the reason is not explained. Figure 2 leaves the reader to figure out which scale (left or right) reads for the temperature and which for the rainfall. Several parameters in table 5 have an ‘asterisk’, probably to indicate that they are significant, but this is not made clear. Errors are provided in table 1 but not for the comparable data in table 4. This means the reader cannot judge the variability in the provided data. In table 1 a particular sample is indicated either positive or negative for a given parameter, no indication which sample is positive and which negative.

Discussion: Appropriate

Bibliography: All references in text and bibliography except for those noted in table 2.

6. PLOS authors have the option to publish the peer review history of their article (what does this mean?). If published, this will include your full peer review and any attached files.

Reviewer #1: No

---

## [Author Response · Author response to Decision Letter 0]

31 May 2021

Reviewer #1: Manuscript No.: PONE-D-20-34257

Title: Impact of heavy precipitation on pathogen occurrence in estuary with oyster farming

Authors: YI-JIA SHIH et al

Overview: The paper investigates the impact of heavy rainfall events on the bacterial and viral load in an estuary. The authors sampled at 8 different points in the estuary. Seasonal samples were collected on a roughly 3 month cycle presumably to provide a baseline. Additional sampling was conducted after a single heavy rainfall event (>80mm/24 hours) and a single very heavy rainfall event (>200mm/24 hours). Water samples were screened for basic chemistry and pathogen load, both bacterial and viral. The authors report variations in pathogenic Vibrio sp., and both adenovirus and norovirus.

General comments: While the study is interesting there appear to be a number of flaws in the experimental design. The data presented is an average of 8 relatively dispersed sample sites. There is no attempt to show how each of the sites varies. Since some of the sites are a kilometer up the estuary while others are offshore, there may be substantial variation in water chemistry and pathogen load that is not being reported. There is no discussion of the hydrography/oceanography of the sample area. Some of the errors indicate substantial variability on the sample data.

Response:

We thank the reviewer for this comment. We have added the variation in average water quality parameters of each sampling site in the supplementary material. Moreover, we have added the environmental characteristics of the Puzi River basin in the discussion section. The relevant sentence has been rewritten as follows: As inshore farming of oysters around the Puzi River estuary is the primary source of oysters in Taiwan, and human activities and livestock farms are located around around the Puzi River basin [48], the wastewater from the river is an important contributor to the water quality of the estuary. Therefore it is necessary to monitor the dynamics of microbial communities in these areas.

Comment:

Title: The title is misleading. The inclusion of oyster farming might suggest that the authors investigated pathogen load in oysters, which they did not. In fact, the relevance to oyster farming is somewhat tenuous. I strongly recommend the reference to an oyster farming region is removed.

Response:

We thank the reviewer for this comment. We have renamed the manuscript as follows: “Impact of heavy precipitation events on pathogen occurrence in estuarine areas of the Puzi River in Taiwan.””

Comment:

Abstract: A little long, but relevant.

Response:

 We have rephrased the Abstract as suggested.

Comment:

Introduction: Appropriate The references to the previous work (bibliography number 27 and 28) of the authors on line 67-69 should be cited with this sentence.

Response:

We have rephrased the sentence on lines 67-69 as followed ”Recently, our group has demonstrated that seasonal changes affect the detection rates and concentrations of human adenovirus in aquatic environments, which is associated with human gastroenteritis. Evidence from previous studies also suggest that the physical determinants and chemical compositions of water bodies might vary seasonally and, consequently, affect microbial communities [27, 28].”

Comment:

Methods: The methodology is somewhat confusing. There are a number of issues that need to be clarified.

1. The locations of the sampling points is somewhat confusing in relation to the data collected. The first section indicates samples were collected adjacent to oyster farms. But no further information is given. Which samples? Where are the oyster farms?

Response:

We thank the reviewer for this suggestion. We have added the description in the Figure 1 caption and illustrated the different types of sampling sites as follows: “Figure 1. Sampling locations around the estuarine areas of the Puzi River in Taiwan Sites A to B were downstream of the Puzi River, sites C to E were fishing ports, and sites F to H were offshore oyster farms. The geographical coordinates (latitude , longitude) are as follows: A (23.458433, 120.177433), B (23.445306, 120.165667), C (23.452656, 120.138769), D (23.450560, 120.139850), E (23.451032, 120.137538), F (23.473775, 120.120292), G (23.444178, 120.132592), and H (23.410706, 120.120975).”

Comment:

2. The first section indicates that ‘seasonal’ samples were collected quarterly. However this is untrue according to the presented data. Seasonal samples were collected in February, May, July and November. This is not quarterly.

Response:

Sampling occurred once per season in this study. We also defined December to February as the winter season. We have added this description as follows: “Moreover, sample collection was also carried out quarterly once per season from winter (December to February was defined as the winter season) in 2016 to winter in 2017 to monitor seasonal changes in water quality background.”

Comment:

3. The water quality data and pathogen load data seems to be a mean of all 8 sample points. These points are reasonably distributed both in estuary and the surrounding coast. Some of the sample point appear to be 2 or 3 kilometers offshore. Is it reasonable to average all these points? Some of the data points have large error.

Response:

In this study, we would like to understand the variation in overall water quality parameters, and these two sampling sites were located downstream of the Puzi River near the estuary; thus, we analyzed them together. However, we have also added the variation in average water quality parameter variation at different sampling sites in the supplementary material.

Comment:

4. Day 1 sampling for the ‘heavy rain’ even is an average of only 5 samples. There is no explanation or indication which sites were excluded.

Response:

Only five samples were collected downstream of the Puzi River and at fishing port sites after the first heavy rain event because of limited sampling from the oyster farm sites. This paragraph has been rewritten as follows:

A total of eight sites were investigated during periods of precipitation events; however, only five samples were collected during rain events. The collection of three offshore samples were banned by the Taiwan Coast Guard Administration for safety reasons. Moreover, sample collection was also carried out quarterly once per season from winter (December to February was defined as the winter season) in 2016 to winter in 2017 to monitor seasonal changes in water quality background. The longitude and latitude for each sampling site are illustrated in Fig 1.

Comment:

5. References listed in table 2 and not in the same format as the rest of the manuscript and are NOT listed in the bibliography.

Response:

We have corrected the format and added the missing reference to the bibliography. 

Comment:

6. I would expect the seasonal sampling to have run for a full 12 months, or more, to ensure that a complete seasonal cycle is evaluated. That would mean that 5 sample sets should have been collected, including a February set in year 2.

Response:

We have added the water quality parameters in February 2017 in Table 1.

Comment:

Results: Reasonably appropriate. A number of the tables and figures need to have more descriptive titles and legends. For instance in table 3, both Vibrio parahaemolyticus and adenovirus have an ‘asterisk’ but the reason is not explained. Figure 2 leaves the reader to figure out which scale (left or right) reads for the temperature and which for the rainfall. Several parameters in table 5 have an ‘asterisk’, probably to indicate that they are significant, but this is not made clear. Errors are provided in table 1 but not for the comparable data in table 4. This means the reader cannot judge the variability in the provided data. In table 1 a particular sample is indicated either positive or negative for a given parameter, no indication which sample is positive and which negative.

Response:

We have added the explanations of the meaning o ‘asterisks’ in the tables, and have added the stranded error in Table 4.

Comment:

Discussion: Appropriate

Response:

 We thank the reviewer for this affirmation.

Comment：

Bibliography: All references in text and bibliography except for those noted in table 2.

Response:

We have rechecked and updated the references lists in the bibliography.

---

## [Decision Letter · Decision Letter 1]

4 Aug 2021

Impact of heavy precipitation events on pathogen occurrence in estuarine areas of the Puzi River in Taiwan

PONE-D-20-34257R1

Dear Dr. Hsu,

We’re pleased to inform you that your manuscript has been judged scientifically suitable for publication and will be formally accepted for publication once it meets all outstanding technical requirements.

Kind regards,

Brett Austin Froelich, Ph.D

Academic Editor

PLOS ONE

Reviewers' comments:

Reviewer's Responses to Questions

**Comments to the Author**

1. If the authors have adequately addressed your comments raised in a previous round of review and you feel that this manuscript is now acceptable for publication, you may indicate that here to bypass the “Comments to the Author” section, enter your conflict of interest statement in the “Confidential to Editor” section, and submit your "Accept" recommendation.

Reviewer #1: All comments have been addressed

2. Is the manuscript technically sound, and do the data support the conclusions?

Reviewer #1: Yes

3. Has the statistical analysis been performed appropriately and rigorously? 

Reviewer #1: Yes

4. Have the authors made all data underlying the findings in their manuscript fully available?

Reviewer #1: Yes

5. Is the manuscript presented in an intelligible fashion and written in standard English?

Reviewer #1: Yes

6. Review Comments to the Author

Reviewer #1: (No Response)

7. PLOS authors have the option to publish the peer review history of their article (what does this mean?). If published, this will include your full peer review and any attached files.

Reviewer #1: No

---

## [Editor Report · Acceptance letter]

6 Aug 2021

PONE-D-20-34257R1 

Impact of heavy precipitation events on pathogen occurrence in estuarine areas of the Puzi River in Taiwan 

Dear Dr. Hsu:

I'm pleased to inform you that your manuscript has been deemed suitable for publication in PLOS ONE. Congratulations! Your manuscript is now with our production department. 

Kind regards, 

on behalf of

Dr. Brett Austin Froelich 

Academic Editor

PLOS ONE